# Diabetes service decentralization to primary healthcare unit in Tigray, Ethiopia: A pilot study

**Merhawit Atsbha Abera**[1], **Haregeweyni Gebreselassie Alemu**[2],
**Hailemariam Berhe Kahsay**[1], **Abraha Hailu Weldegerima**[1], **Afework Mulugeta**[1],
**Mengistu Hagazi Tequare**[1], **Afewerki Tesfahunegn Nigusse**[1]*,
**Migbnesh Gebremedhin Weledegebriel**[1], **Tsega Cherkos Dawit**[1], **Ephrem Berhe**[1],
**Tarekegn Geberhiwot**[3]

**1** College of Health Sciences and Ayder Comprehensive Specialized Hospital, Mekelle University, Mekelle, Ethiopia, **2** Institute for Health Care Improvement, Mekelle, Ethiopia, **3** University Hospital of Birmingham NHS foundation Trust, Birmingham, United Kingdom

* afom.te@gmail.com

## Abstract

### Background

The prevalence of diabetes mellitus has been increasing in the past few decades. Mortality and morbidity have increased faster in low- and middle-income countries than in high-income countries. In Ethiopia it has been a practice to handle most diabetes mellitus patients in general and referral hospitals which in turn caused over burden in these hospitals. Besides, the possibility of implementing diabetic care service in primary health care facilities is not assessed in Ethiopia. The aim of this study was to decentralize and implement diabetic service in primary health care facilities in Tigray.

### Methods and materials

The study was conducted in Tigray region, at Hagereselam primary hospital from September 2019 to September 2020. A pilot study which aimed to bring diabetes services to non-specialized health care facilities. Diabetic patients who were on follow up in tertiary hospital (Ayder Comprehensive Specialized Hospital) were moved to Hagereseam primary hospital voluntarily, which is the main study site. The data were collected by trained health professionals and the data collection tool was adapted and developed from national guideline and analysed through SPSS 21. Independent t-test and chi-square was applied to compare the outcomes among study participant groups.

⚲ OPEN ACCESS

**Data availability statement:** The datasets used and/or analysed during this study are attached with this submission as supplementary file.

**Funding:** This study is funded by Mekelle University for material procurement and personal costs related to data collection and follow-up. The fund was received Dr Merhawit Atsibeha Abera from Mekelle University recurrent budget in 2019 with the reference number of mu/chs/ ERC 1352/2019. The funder website is https://www.mu.edu.et/ the funder didn't have any role on the design of the study.

**Competing interests:** The authors have declared that no competing interests exist.

**Abbreviations:** ACSH, Ayder Comprehensive Specialized Hospital; DM, Diabetes Mellites; FBS, Fast Blood Sugar; HAART, Highly Active Antiretroviral Therapy; HCPs, Health Care Providers; IQR, Inter Quartile Range; ODK, Open Data Kit; PHCU: Primary Health Care Unit.

## Results

The mean and median Fasting Blood Sugar (FBS) level was 171mg/dl and 151.5 (IQR = 109.5–180.7) respectively. and the mean difference of FBS with randomly selected patients in a referral hospital was 3.55 (p = 0.8) which shows no significant difference. Mean systolic BP and haemoglobin A1c were 114.2 mmHg and 8.34% respectively.

## Conclusion

There is no significant difference in the diabetes service among the primary and tertiary hospital. Diabetes service can be decentralized to primary health care facilities without compromising the quality of diabetic care. Further large implementational study is necessary to overcome the problems in the decentralization of service delivery in diabetes service.

## Introduction

Diabetes Mellitus (DM) is a chronic condition that occurs when the body cannot produce enough insulin or cannot use insulin efficiently. It is the leading cause of end stage renal failure and blindness and the second leading cause of amputation next to trauma [1,2]. Globally, it is estimated that 537 million adults in the age range of 20 and 79 have diabetes mellitus making the prevalence 10.5%. In 2021 International diabetes federation has estimated that one in ten people have diabetes, one in two adults with diabetes is undiagnosed, one in twelve births are affected by diabetes and 966 billion dollar of global health expenditure is spent on diabetes [1].

The prevalence of diabetes has been steadily increasing for the past 3 decades and is growing most rapidly in low- and middle-income countries. In Africa more than 24 million adults are estimated to live with diabetes making the prevalence 4.5% and 54% of people living with diabetes are undiagnosed [1].

Ethiopia has 2.57 million adults with diabetes (5.2% of adult population) and 4.9 million adults with pre diabetes [1]. This is estimated to be the largest diabetes population in Sub-Saharan Africa, while 76% of the population do not even know that they have diabetes. Thirty percent of deaths are due to non-communicable diseases (NCD) and among these diabetes accounts 2.6% [1,3].

In Ethiopia it is a trend to handle most diabetes mellitus patients in general and referral hospitals. This has caused over burden and has made these hospitals unable to focus their resources in more complicated cases of diabetes. Due to the high number of patients seen in the diabetes clinics, patients are not even having proper screening of chronic complications of diabetes. It also has impact on patients; most patients who come from the country side are having extra cost for their transportation and accommodation among loss of earnings [4].

Studies conducted in Brazil, Philippians, and US have shown that decentralized diabetes service has an impact on diabetes morbidity and mortality. Decentralisation

of the service has a good long term and short-term outcomes, where, no significant difference was observed on necessary diabetic tests, risk factors assessments and service delivery cares among referral and primary care health facilities [5–7]. Therefore, the aim of this pilot study was to decentralize diabetes care to primary health care facilities and compare the implementation with the diabetic care services given in tertiary hospital.

## Methods and materials

### Study setting and design

A pilot study was conducted in Hagere Selam primary hospital and catchment health centers. Hagereselam primary hospital is located in Hagereselam town 50 Km away to the west of Mekelle, the capital city of Tigray. Hagere Selam is located at 13°39′N 39°10′E with an elevation of 2650 metres above sea level. The primary hospital served for a total population of 172,848 and have five health centers under the district [8]. The study was implemented from September 2019 to September 2020.

Training for health care professionals and assessment of the health care facilities was done in the first 3 months. The health care professionals (HCPs) in the health centres were trained on how to pick newly diagnosed patients and acute complications of diabetes and make a timely referral to the primary hospital. While the health care professionals in the primary hospital were trained on the acute and chronic complications of diabetes, titration of commonly available diabetes medications and timely referral of patients who cannot be managed in the primary hospital.

### Population

Diabetic patients resided in Hagere Selam catchment area (Degua-Tembien district) and had follow up in the ACSH were the study population. Patients with diabetes in Hagere Selam catchment area who had followed up in Ayder comprehensive specialized hospital and willing to move their follow-up to Hagere-Selam primary hospital were taken as sampling unit. Voluntary patients who were willing to continue their follow up in Hagere-Selam primary hospital were recruited from the diabetic clinic of Ayder comprehensive Specialized Hospital (ACSH) after informing all necessary information regarding their follow-up in the intervention area (Hagere-Selam Primary Hospital). All the patients moved from ACSH to Hagere-Selam primarily hospital on volunteer basis were included in the study conveniently. A total of 63 DM patients from the total 350 patients who had follow up in ACSH and willing to go to Hagere Selam primary hospital were moved to the implementation site on December 2019 and followed their status and service delivery of the primary hospital regularly by physicians and public health experts. A total of 67 randomly selected DM patients who have follow up in ACSH and comes from similar set-up of the transferred patients to the implementation site were included to look and compare the quality of DM services, outcomes, and clinical parameters.

### Interventions

**Health facility assessment.** Before the intervention, a baseline structured health facility assessment was conducted to assess the readiness of the health facilities using the World health organisation service availability and assessment (SARA) tool; to identify the available resources and the gaps to be filled in order to provide minimum standardized care for diabetic patients [9]. Lack of basic health resources for the screening, diagnosis and treatment of diabetes mellitus was identified from the primary health facility readiness assessment. Accordingly, the necessary materials and equipment were supplied before starting the intervention.

**Training of health care providers (HCPs).** All Health care professionals (general practitioners, health officers, nurses, pharmacists and lab technicians) in the selected PHCUs were trained on diabetes mellitus (DM) identification and management by experienced internists and master trainers trained by Ethiopian federal ministry of health. The general practitioners, health officers and nurses were trained to do the diagnosis, treatment, follow up and to provide advices to

patients and families. Pharmacists and lab technicians were trained on supply chain management of drugs and supplies. The content of the training included presentations, group discussions, and practical sessions in the diabetic clinic. Management, diagnosis and referral algorithms were developed and provided to the trainees to have clear approach and referral system. All HCPs undergone standardization exercises for identification of DM patients. The training included assessment of 5–10 DM patients in a community and nearby health facility. Refreshment trainings and six-monthly re-standardization sessions were conducted for all HCPs during the study implementation period. Besides, regular supervisory supports were conducted by the staff from internal medicine and public health departments of ACSH.

**Follow-up and referral of patients.** All DM patients under follow-up in the PHCU were evaluated at days 30, 60 and 90 by the same treating HCPs. DM patients with disease progression were advised for referral. The health worker facilitated referral by counselling about things that need to be done on the way to the referral hospital and providing a referral slip. On the regular follow-up visits, the patients were examined for signs of improvement or deterioration and for any adverse effects. If the condition of the patient deteriorates at any visit, the patients were referred to ACSH again for further assessment and treatment as clinically indicated. If the patient refuses to accept hospital referral despite the best efforts of the HCPs at the health facility, they were offered treatment at the PHCU.

## Data collection

A structured data collection tool was prepared and adapted from literatures by the investigators [5–7,10–13]. The data collection tool includes socio-demographic, behavioural and comorbidity characteristics, clinical symptoms and measurements, and treatments and complication. Data were collected from the patients in both group by the prepared questionnaire through ODK and the quality was maintained by trained the data collectors and strictly followed the process by the protocol. Quality of diabetic care service was measured by variables such as fasting blood sugar (FBS) and haemoglobin A1C. Baseline data and end stage data were collected from September 2019 up to September 2020 from the patients both orally and using laboratory samples to assess the effectiveness of the interventions and compare with the control site. Sociodemographic variables are considered as predictors of the outcome variables. Service satisfaction related data was collected from the implementation site.

## Data analysis

Data were extracted from the server to csv file then analysed by SPSS 21. Descriptive data was analysed using frequency table and compared with chi square. Mean and standard deviation was computed for continuous data for those normally distributed data and median and interquartile range for those skewed data. Selected indicators for both groups were compared and analysed using independent t-test for continuous data and Chi square for categorical data in both hospitals. Follow up data were compared using paired t-test in both group of participants. Assumptions for both t-test and chi square were checked and the data fulfilled the criteria.

## Ethics approval

Ethical clearance was obtained from Institutional review board of Mekelle University College of health science (ERC 1352/2019). Permission letter was obtained from Tigray regional health bureau and Ayder comprehensive specialized hospital. Written consent was obtained from each participant and confidentiality was kept seriously. All the process was done based on the ethical declaration of Helsinki declaration.

## Result

### Socio-demographic characteristic of participants

In the follow-up period, a total of 67 participants responded to the study. The mean age of the participants was 42 (±16.7). Thirty-five (52.2%) participants were male. Out of all participants, 60 (89.5%) of them had their service care for free, which was

paid by the regional government and the rest were paying for the service. Forty-eight (71.6%) were members of community-based health insurance. The average time that takes to travel between home and the hospital was 2.3 (±5.4) hours (Table 1).

## Behavioural and comorbidity characteristics

A total of four (5.97%) participants were newly diagnosed diabetic patients with one had family history, and the rest were previously diagnosed patients. From the total 34 (50.8%) participants were DM type-2 and the median and interquartile range of patients on DM follow up was 4 (4.83) years. Twenty (29.9%) and one (1.5%) of the participants were local alcohol drinkers (Sewa and Areki) and cigarette smokers, respectively. Fifty-nine (88.1%) eat their meal three times per day. Around 27% of them were regular table sugar users; they used mainly to add with coffee and for treatment of hypoglycaemia. The mean amount of water intake per day was 1.1 (±0.64) litres. The number of participants who already have received diabetic education based on specific topics were: diet (100%), exercise (85%), medication (100%), acute complication (44.8%), chronic complication (17.9%), and foot ulcer (44.8%)). None of the participants missed their work or school due to their illness in the study period. Comorbidity incudes; one participant had thyroid disease, one history of tuberculosis, one vitiligo, and one had HIV/AIDS and was on HAART (Table 2).

## Clinical symptoms and measurements

Regarding the clinical symptoms of DM, 19 (28.2%) and 14 (20.9%) of the participants had poly symptoms and visual disturbance respectively during the study period. Seventeen (25.3%) of the participants had peripheral nerve symptoms (numbness (2), paraesthesia (13), hyper paraesthesia (03), and pain (04)). One of the participants had localized extremity skin discoloration. Dorsalis pedis pulse 56, 07, and 01 are intact, feeble, and absent for right side (Table 3).

The mean Haemoglobin A1C level after 3 months of follow up in Hagereselam was 8.34% Which was similar to the one taken in Ayder (8.97%) in 67 randomly selected patients. The mean Serum creatinine level was 0.71 mg/dl which was similar with the patients in Ayder (0.73 mg/dl). No significant difference was observed in terms of Hemoglobin A1C

**Table 1. Socio-demographic characteristics of Diabetic patients in the pilot study site, 2020.**

| Variable | Category | No (%) |
|---|---|---|
| Age | <30 years | 22 (32.8%) |
| | ≥30 years | 45 (67.2%) |
| Sex | Male | 35 (52.2%) |
| | Female | 32 (47.8%) |
| Marital status | Single | 8 (11.9%) |
| | Married | 47 (70.1%) |
| | Widowed and divorced | 12 (18%) |
| Educational status | No formally attend school | 38 (56.7%) |
| | formally attend school | 29 (43.3%) |
| Occupation | Farmer | 30 (46.9%) |
| | House wife | 19 (29.7%) |
| | Employee | 11 (17.2%) |
| | Student | 4 (6.2%) |
| Members of DM association | Yes | 36 (54.5%) |
| | No | 30 (45.5%) |
| Mode of travel | Foot | 41 (61.2%) |
| | Car | 14 (21%) |
| | Other | 12 (17.8%) |

**Table 2. Behavioural and co-morbidity status of diabetic patients in pilot study site, 2020.**

| Variable | Category | No (%) |
|---|---|---|
| Type of DM | Type 1 | 33 (49.2%) |
| | Type 2 | 34 (50.8%) |
| Alcohol drinking | Yes | 4 (4.8) |
| | No | 63 (95.2) |
| Cigarette smoking | Yes | 1 (1.5) |
| | No | 66 (98.5) |
| Regular sugar user | Yes | 18 (27) |
| | No | 49 (73) |
| Received education | Diet | 67 (100) |
| | Exercise | 57 (85) |
| | Medication | 67 (100) |
| | Acute complication | 30 (44.8) |
| | Chronic complication | 12 (17.9) |
| | Foot ulcer | 30 (44.8) |
| Number of meals per day | ≤2 | 5 (7.6%) |
| | ≥3 | 61 (92.4%) |
| Received DM education booklet | Yes | 2 (3.2%) |
| | No | 60 (96.8%) |
| Regular exercise | Yes | 39 (58.2%) |
| | No | 28 (41.8%) |
| Hypertension | Yes | 7 (10.9%) |
| | No | 57 (89.1%) |

**Table 3. Clinical symptoms and measurements of diabetic patients in pilot study site, 2020.**

| Variables | Category | Frequency (%) |
|---|---|---|
| Poly symptoms | Yes | 19 (28.2) |
| | No | 48 (71.8) |
| visual disturbance | Yes | 14 (20.9) |
| | No | 53 (79.1) |
| Peripheral nerve symptoms | Yes | 17 (25.3) |
| | No | 50 (74.7) |
| Right dorsalis pedis pulse | Intact | 56 (87.5) |
| | Feeble | 07 (10.9) |
| | Absent | 01 (1.6) |
| Left dorsalis pedis pulse | Intact | 55 (85.9) |
| | Feeble | 08 (12.5) |
| | Absent | 01 (1.6) |

and creatinine, with P value of 0.104 and 0.506 respectively (Table 4). Of the total participants in the implementation site 94.1% were satisfied by the decentralization of the service.

## Treatment and complication

Ten (14.9%) experienced hypoglycaemias with the mean of episode per month being 3.67(±1.73). This was similar with hypoglycaemia in randomly selected patients in ACSH 19 (28.4%) with mean episodes per month of 2.5(±1.13). Of those

**Table 4. Clinical parameters of diabetic patients in Hagereselam primary hospital, 2020.**

| Variables | Mean | Median (IQR) |
|---|---|---|
| Fasting blood sugar | 171mg/dl | 151.5 (109.5–180.7) mg/dl |
| Systolic blood pressure | 114.2mmHg | 111 (100–127) mmHg |
| Diastolic blood pressure | 72.7 mmHg | 74 (69–79) mmHg |
| Haemoglobin A1C | 8.34% | 8 (6.8–10.1) % |

who faced hypoglycaemia eight of them were in pre hospital area and the rest were in both hospital and pre hospital with the blood sugar level of 56mg/dl and 68mg/dl. Both of the hypoglycaemic participants who arrived in the hospital were conscious and one of them received oral 40% glucose and medication was tapered after his sugar level normalized. None of the participants discontinued their DM medication and only one participant reported sever dyspepsia attributed to the treatment.

## Discussion

The main goal of this pilot study was to evaluate decentralized diabetic care service in primary health care facilities (primary hospitals). The intention was to provide diabetic care services close to the patient home with a comparable level of care to the specialised health facilities. By doing so patients can get access to diabetic care in nearby health facilities and the capacity in specialised health care facilities can be optimized. Before starting the intervention, structured facility readiness assessment was done and the required resources were fulfilled according to the standard [9].

In this study, diabetes service delivery in primary health care unit has helpful to the patients where no significant difference was observed in their clinical parameters. This is also supported by the patient satisfaction level in which more than 94% participants were satisfied with the service. This is supported with similar studies conducted in Ethiopia and Rwanda on NCD and favoured decentralization of care to non-specialist health care facilities. For instance, a study in Rwanda showed nurse-led outpatient NCD programme to all first-level hospitals with good fidelity, feasibility and penetration as to expand access to care for severe NCDs [10–12]. This could be due to less patient load in the primary cares and got time for detail follow-up assessments and on time and strict medication follow-ups.

The study conducted in Malawi showed that, gender has an effect on diabetes effective follow-up and diabetic control, but in our study, gender of the participants has no effect on the diabetes follow-up and complication development [13]. This could be due to the difference in study subjects, sizes of the sample and study design type.

There was no significant difference on the mean Haemoglobin and creatinine results between the patients in Hagereselam primary hospital and randomly selected similar number of patients in ACSH. This finding is similar with the study conducted in various place of the globe within and outside Africa. The study conducted in Rwanda, South Africa, Malawi, Ethiopia, Philippines, and Brazil [5–7,10–13] supported the result of this study. In this study, standard checklist-based assessment was conducted before the implementation and done all procedures accordingly. This result and similarities could be due to various factors such as accessibility and affordability of services in nearby facility reduce the stress of the patients, having good patient-provider interaction during counselling due to less patient follow-up, and similar quality of services.

The finding of this study is supported with studies conducted in LMIC and high-income countries [5–7,10–13], therefore, by decentralizing diabetes care to primary health care facilities service with the same quality could be provided with limited budget. It will also decrease the transportation and accommodation expenses of patients as it helps them get the service in nearby health facilities. On the other hand, this will help the general and referral hospitals to focus on more severe and complicated cases.

## Limitation

Initially the intervention was designed to be implemented for extended period of time (3 years) beyond this pilot study that was divided in to different phases of follow up to get adequate number of patients so as to maintain validity of the study.

Unfortunately, the study was interrupted after conducting the pilot study owing to the Tigray war in November 2020. As a result, the current finding is not the whole package of the study as the then planned but a fraction of it (the sample size is small and the method of the participants selection). Hence, this study has its own limitation in terms of generalizing its findings to a bigger community.

## Conclusion

From this study it can be concluded that diabetic care services can be decentralized to non-specialized facilities such as primary hospitals as quality of diabetic care (explained by FBS, systolic blood pressure, Hemoglobin A1C and creatinine) will not be compromised. Further large implementational study is necessary to overcome the problems in the decentralization of service delivery in diabetes service.

## Supporting information

**S1 File. Manuscript data.**
(XLSX)

## Acknowledgments

We would like to thank our study participants for their willingness, our data collectors, and the hospital staff of both Ayder Comprehensive and Specialized Hospital and Hagere-Selam Primary Hospital.

## Author contributions

**Conceptualization:** Merhawit Atsbha Abera, Haregeweyni Gebreselassie Alemu, Afework Mulugeta, Migbnesh Gebremedhin Weledegebriel.

**Data curation:** Afewerki Tesfahunegn Nigusse, Tarekegn Geberhiwot.

**Formal analysis:** Afewerki Tesfahunegn Nigusse.

**Funding acquisition:** Merhawit Atsbha Abera.

**Investigation:** Merhawit Atsbha Abera, Hailemariam Berhe Kahsay, Mengistu Hagazi Tequare, Tsega Cherkos Dawit.

**Methodology:** Merhawit Atsbha Abera, Abraha Hailu Weldegerima, Mengistu Hagazi Tequare, Afewerki Tesfahunegn Nigusse, Migbnesh Gebremedhin Weledegebriel.

**Project administration:** Afework Mulugeta.

**Supervision:** Merhawit Atsbha Abera, Haregeweyni Gebreselassie Alemu, Hailemariam Berhe Kahsay, Abraha Hailu Weldegerima, Mengistu Hagazi Tequare, Migbnesh Gebremedhin Weledegebriel, Tsega Cherkos Dawit, Ephrem Berhe.

**Validation:** Hailemariam Berhe Kahsay, Afework Mulugeta, Tarekegn Geberhiwot.

**Visualization:** Ephrem Berhe.

**Writing – original draft:** Merhawit Atsbha Abera, Afewerki Tesfahunegn Nigusse.

**Writing – review & editing:** Merhawit Atsbha Abera, Haregeweyni Gebreselassie Alemu, Hailemariam Berhe Kahsay, Abraha Hailu Weldegerima, Afework Mulugeta, Mengistu Hagazi Tequare, Afewerki Tesfahunegn Nigusse, Migbnesh Gebremedhin Weledegebriel, Tsega Cherkos Dawit, Ephrem Berhe, Tarekegn Geberhiwot.

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
