## [Decision Letter · Decision Letter 0]

29 Nov 2024

PONE-D-24-29698Decentralising diabetes care from tertiary hospital to primary health care units in Tigray, Ethiopia: A pilot studyPLOS ONE

Dear Dr. Nigusse,

Thank you for submitting your manuscript to PLOS ONE. After careful consideration, we feel that it has merit but does not fully meet PLOS ONE’s publication criteria as it currently stands. Therefore, we invite you to submit a revised version of the manuscript that addresses the points raised during the review process.

We look forward to receiving your revised manuscript.

Kind regards,

Efrem Kentiba, PhD

Academic Editor

PLOS ONE

**Journal Requirements:**

We would like to thank our study participants for their willingness, our data collectors, and the hospital staff. We would like to acknowledge Mekelle University, college of Health Sciences for funding and creating the opportunity for us to be involved in research projects.

This study is funded by Mekelle University for material procurement and personal costs related to data collection and follow-up. The fund was received Dr Merhawit Atsibeha Abera from Mekelle University recurrent budget in 2019 with the reference number of ERC 1756/2020. The funder website is https://www.mu.edu.et/

the funder didn't have any role on the design of the study.

4. In the online submission form, you indicated that The datasets used and/or analysed during this study are available from the corresponding author on reasonable request.

Reviewers' comments:

Reviewer's Responses to Questions

**Comments to the Author**

1. Is the manuscript technically sound, and do the data support the conclusions?

Reviewer #1: Partly

Reviewer #2: Partly

Reviewer #3: Yes

2. Has the statistical analysis been performed appropriately and rigorously? 

Reviewer #1: Yes

Reviewer #2: Yes

Reviewer #3: Yes

3. Have the authors made all data underlying the findings in their manuscript fully available?

Reviewer #1: Yes

Reviewer #2: No

Reviewer #3: No

4. Is the manuscript presented in an intelligible fashion and written in standard English?

Reviewer #1: Yes

Reviewer #2: Yes

Reviewer #3: Yes

5. Review Comments to the Author

**Reviewer #1:**  Dear Authors,

I want to commend you on the excellent work you’ve done. Your research has the potential to significantly impact diabetes mellitus (DM) care in Tigray, which is both timely and necessary. However, I have several concerns regarding the manuscript that I believe need to be addressed to strengthen your work further.

I hope these suggestions will be helpful as you revise the manuscript. I look forward to seeing the final version and am confident that, with these adjustments, your work will be even more impactful.

Thank you once again for your contributions.

Title: Diabetes service decentralize to Primary health care unit; A pilot study

Short title

Healthcare unit, not health case unit.

Abstract

Introduction

“Ethiopia has 2.57 million adults with diabetes (5.2 % of adult population) and 4.9 million adults with prediabetes. This is estimated to be the largest diabetes population in Sub-Saharan Africa. 76% do not even know that they have diabetes.” Please put references for both statements.

“Studies have shown that decentralized diabetes care has effect of diabetes morbidity and mortality and has good long term and short-term outcome. In addition, there was no significant difference in conducting necessary diabetes test, risk factors assessments and service delivery cares among referral and primary care health facilities (4-6).”

This is not clear. Which studies? Which countries or settings? Please rewrite this section.

Study setting

- It is not clear about the patients the authors moved from Ayder to Hagereselam. The authors did not explain, for example, if the patients moved from Ayder are diabetic patients who are residents of Hagereselam and its vicinity.

- Please mention Hagereselam’s elevation, demographics etc.

- Please describe the steps you followed to discuss with patients to transfer them from Ayder to Hagereselam. How many patients have you contacted to get 67 willing patients? How many patients from Hagereselam’s vicinity had follow up in Ayder? How many agreed to be moved to Hagereselam? How did you address their concerns? Unless you explain these things, we cannot know if the patients were moved voluntarily or coerced.

Results

- In the study setting you mentioned that newly diagnosed patients from Hagereselam and veteran patients from Ayder Hospital were the study population. You implied that for this purpose, healthcare professionals were trained to diagnose new diabetic patients. However, in the results section, you mentioned that only those transferred from Ayder were followed. Please clarify this.

- It is not clear how you involve patients from Ayder for comparison. You have not mentioned in your methods section about patients selected in Ayder for comparison. However, in your results section, you mentioned that you compared your 3 months lab results with randomly selected patients. This needs an explanation in your methods section. Who is this patient population you are comparing with your study population? Are you talking about the same patients you are using for pilot, or you used the same patients’ prior records? If the case is the later, it is not right as you are neglecting temporal effects.

Discussion

- You are implying that patients are satisfied. How did you assess their satisfaction levels? What model have you used to measure it?

Limitations

- Why did the authors decide to cut short/abort the study? It would have made sense to continue it to get a good picture even after the Cessation of Hostilities agreement. As they have clearly stated, the study does not show the whole picture. For example, though not very clearly stated, they mainly relied on veteran patients from Ayder to conclude. Based on this study’s findings, we cannot conclude that new patients are safe to be treated at the primary healthcare units.

**Reviewer #2: ** The study explores the decentralization of diabetes care in Ethiopia, which is an under-researched area in the context of sub-Saharan Africa. However, similar studies have been conducted in other low- and middle-income countries (LMICs), such as Rwanda, Malawi, and the Philippines, as cited in the manuscript. The context-specific pilot study in Tigray, Ethiopia, adds a localized perspective, addressing challenges like hospital overcrowding and accessibility in rural areas. This is a valuable contribution to Ethiopian healthcare policy discussions. There is no novelty in the study. The concept of decentralizing diabetes care is not groundbreaking, and the study primarily replicates approaches tested in other LMICs. The study would benefit from emphasizing unique findings or challenges specific to Ethiopia’s healthcare system.

The manuscript lacks an in-depth exploration of the systemic barriers and facilitators to decentralization, such as infrastructure, cultural perceptions, or financial constraints. The study does not sufficiently discuss the scalability or sustainability of its findings. For example, how would decentralization fare with larger patient cohorts or in different Ethiopian regions? Small sample size (67 patients in each group) limits the generalizability of findings.

Convenience sampling may introduce bias, as patients willing to transfer may differ systematically from those who remain in tertiary care. The study duration (one year) is not capture long-term trends or complications.

The discussion underemphasizes the challenges faced during implementation, such as resource limitations or patient adherence issues. The comparison with other studies lacks critical depth. For instance, what systemic differences make decentralization more or less effective in Ethiopia compared to other countries? The impact of training on healthcare provider performance is not evaluated quantitatively.

Furthermore, the manuscript does not address how decentralization could impact healthcare equity. For example, are rural areas with limited resources at risk of unequal care quality? The manuscript makes a useful contribution to the understanding of decentralized diabetes care in Ethiopia, but its novelty is limited by the replication of methods tested in other LMICs. The small sample size, lack of long-term data, and underexplored systemic challenges weaken the overall impact. Refining the discussion, emphasizing Ethiopia-specific insights, and addressing methodological limitations will significantly enhance the manuscript’s value.

**Reviewer #3:**  1. In introduction, paragraph no. 3, line no 3, it has been written as "76% do not even know that they have diabetes", it is suggested not to start the sentence with numerical value.

2. There are no references in the fourth paragraph of introduction.

3. In data collection tool: what type of questionnaire was used (structured, semi-structured...), what were the major contents of data collection tools (written but not clear)?

4. Result: (Needs to be re-written)

- "Behavioural and comorbidity characteristics section:, it is suggested to make the description and figure uniform.

- "Clinical symptoms and measurements" section: description has been provided, however, there is no table (Table no. 3 has been mentioned to refer for the result though)

- No figure for comparison related data (only description provided)

5. In consent to participate, it is suggested to replace the word 'clients' with participants. It is better if sentences would be added on anonymity, risk if any.

Confusion:

- Are the both intervention group and comparison group same?? I mean both patients group? May be I have misunderstood.

6. PLOS authors have the option to publish the peer review history of their article (what does this mean? ). If published, this will include your full peer review and any attached files.

**Do you want your identity to be public for this peer review?** For information about this choice, including consent withdrawal, please see our Privacy Policy .

Reviewer #1: **Yes: ** Hale Teka

Reviewer #2: **Yes: ** Junaid Ahmad

Reviewer #3: No

---

## [Author Response · Author response to Decision Letter 1]

17 Jan 2025

Response to Reviewers

Dear Journal Editor (Dr. Efrem Kentiba)

Thank you very much for your timely response of the reviewer’s comments, suggestions and guidance of our manuscript. Here below is the point-by-point response of the reviewer’s query.

The style requirements have been addressed based on the guideline and the information regarding the funding information has been included in the cover letter in order to revise it in the system. Thank you, we have revised the acknowledgment section based on your comments and the guideline you sent to us. The ethical clearance section has been moved into the methods and materials section, and the data we have used in this study is here attached as a supplementary file in Excel file type.

Reviewer #1 (Hale Teka)

Title: Diabetes service decentralize to Primary health care unit; A pilot study

Response: Thank you very much, we have revised accordingly. (line 1-2)

Introduction

1. “Ethiopia has 2.57 million adults with diabetes (5.2 % of adult population) and 4.9 million adults with prediabetes. This is estimated to be the largest diabetes population in Sub-Saharan Africa. 76% do not even know that they have diabetes.” Please put references for both statements.

Response: Thank you for your suggestion, we have revised accordingly. (line 58-62)

2. “Studies have shown that decentralized diabetes care has effect of diabetes morbidity and mortality and has good long term and short-term outcome. In addition, there was no significant difference in conducting necessary diabetes test, risk factors assessments and service delivery cares among referral and primary care health facilities (4-6).”

This is not clear. Which studies? Which countries or settings? Please rewrite this section.

Response: Thank you for the question, we have revised and included the study setting of the studies and rephrased the sentence too. (line 69-75)

Study setting

- It is not clear about the patients the authors moved from Ayder to Hagereselam. The authors did not explain, for example, if the patients moved from Ayder are diabetic patients who are residents of Hagereselam and its vicinity.

Response: Thank you for your valuable comment, we have revised the section with its explanation at the population sub-section of the method part. The moved patients were DM patients who agreed to follow their diabetic care at the primary health care health facilities. (line 91-107)

- Please mention Hagereselam’s elevation, demographics etc.

Response: We have revised accordingly (line 80-83)

- Please describe the steps you followed to discuss with patients to transfer them from Ayder to Hagereselam. How many patients have you contacted to get 67 willing patients? How many patients from Hagereselam’s vicinity had follow up in Ayder? How many agreed to be moved to Hagereselam? How did you address their concerns? Unless you explain these things, we cannot know if the patients were moved voluntarily or coerced.

Response: Thank you for your question. We have assessed the number of visits from different parts of Tigray in Ayder comprehensive specialized hospital and we have got 350 visits from Hagereselam and its catchment area. Every patient was asked for their consent and those who are willing were informed to continue follow up in Hagereselam starting from the next possible visit. Referral paper was given from Ayder to make their follow up in Hagereselam smooth. A total of 63 patients were moved and four of them were diagnosed in Hagereselam primary hospital. (line 91-107)

Results

- In the study setting you mentioned that newly diagnosed patients from Hagereselam and veteran patients from Ayder Hospital were the study population. You implied that for this purpose, healthcare professionals were trained to diagnose new diabetic patients. However, in the results section, you mentioned that only those transferred from Ayder were followed. Please clarify this.

Response: Thank you for the suggestion and concern; we have revised accordingly. Most of the participants (94.03%) were moved from Ayder Comprehensive Specialized Hospital for this study’s purpose, but we have four newly diagnosed participants in the implementation area as a result of the training and other implementation activities. (line 100-104 and 179-180)

- It is not clear how you involve patients from Ayder for comparison. You have not mentioned in your methods section about patients selected in Ayder for comparison. However, in your results section, you mentioned that you compared your 3 months lab results with randomly selected patients. This needs an explanation in your methods section. Who is this patient population you are comparing with your study population? Are you talking about the same patients you are using for pilot, or you used the same patients’ prior records? If the case is the later, it is not right as you are neglecting temporal effects.

Response: Thank you for your questions. We have selected a total of 67 participants from Ayder Comprehensive Specialized Hospital to compare the clinical parameters and the outcomes. We have mentioned this in the population subsection of the method part. The randomly selected participants from Ayder were, DM patients who had follow-up visits and came from a similar set-up to the patients transferred to Hagere selam primary hospital. Primary data was collected from the patients and laboratory results from the records of the patients other than the patients transferred to Hagere Selam the implementation site. (line 104-107)

Discussion

- You are implying that patients are satisfied. How did you assess their satisfaction levels? What model have you used to measure it?

Response: A total of 12 questions with a five-point Likert scale score were used to collect the participants satisfaction at the implementation site. Participants who scored above the mean value of the five-point Likert scale were considered as satisfied, while the rest were deemed not satisfied with the DM service. (line 205-206)

Limitations

- Why did the authors decide to cut short/abort the study? It would have made sense to continue it to get a good picture even after the Cessation of Hostilities agreement. As they have clearly stated, the study does not show the whole picture. For example, though not very clearly stated, they mainly relied on veteran patients from Ayder to conclude. Based on this study’s findings, we cannot conclude that new patients are safe to be treated at the primary healthcare units.

Response: Thank you for your suggestions. This pilot study ended in September 2020, and a month later, the war in Tigray broke out, changing everything. The health facilities in the district, including the primary hospital, were destroyed and looted, compromising services in both the primary hospital and the comprehensive specialized hospital. Due to this and other financial constraints, the study did not continue as planned for large - scale implementation. Yes, you are correct; 63 of the 67 participants in the implementation site were transferred from the comprehensive hospital and followed up in the implementation site for approximately one year. We have described this in our limitations section. (line 254-261)

Reviewer #2 (Junaid Ahmad):

• The study explores the decentralization of diabetes care in Ethiopia, which is an under-researched area in the context of sub-Saharan Africa. However, similar studies have been conducted in other low- and middle-income countries (LMICs), such as Rwanda, Malawi, and the Philippines, as cited in the manuscript. The context-specific pilot study in Tigray, Ethiopia, adds a localized perspective, addressing challenges like hospital overcrowding and accessibility in rural areas. This is a valuable contribution to Ethiopian healthcare policy discussions. There is no novelty in the study. The concept of decentralizing diabetes care is not groundbreaking, and the study primarily replicates approaches tested in other LMICs. The study would benefit from emphasizing unique findings or challenges specific to Ethiopia’s healthcare system.

Response: Thank you for your in-depth thought, our aim of this study was to implement the DM service utilization with the proven interventions in another similar set-up. We have addressed the challenges of the studies conducted in LMICs in our study by following the strict intervention activities. Unfortunately, the pilot study was stopped before scaled-up and largely implemented due to the eruption of the humanitarian crisis due to the northern Ethiopia war in November 2020.

• The manuscript lacks an in-depth exploration of the systemic barriers and facilitators to decentralization, such as infrastructure, cultural perceptions, or financial constraints. The study does not sufficiently discuss the scalability or sustainability of its findings. For example, how would decentralization fare with larger patient cohorts or in different Ethiopian regions? Small sample size (67 patients in each group) limits the generalizability of findings. Convenience sampling may introduce bias, as patients willing to transfer may differ systematically from those who remain in tertiary care. The study duration (one year) is not capturing long-term trends or complications.

Response: Thank you for your comment, your concern is valid and had risk of bias (voluntary bias). We have tried to overcome this bias by providing the same and deep counselling methods and also put it under the limitation section as it is difficult to prevent the bias fully. The study was intended to continue until a significant number of patients are transferred and scaled-up to larger implementation sites following the pilot study; but because of the war that was not possible and every material in the new clinic was looted so we could not continue the study because of lack of materials and medications to run it.

• The discussion underemphasizes the challenges faced during implementation, such as resource limitations or patient adherence issues. The comparison with other studies lacks critical depth. For instance, what systemic differences make decentralization more or less effective in Ethiopia compared to other countries? The impact of training on healthcare provider performance is not evaluated quantitatively.

Response: The resources that were important for diabetes care in a primary health care facility were availed based on WHO service availability and readiness assessment tool and strictly followed the procedure. The methods we used were different from that used in other parts of Africa so it was difficult to make direct comparison. Of course, the result of the training was compiled and compared before and after a training and in their health facility and had a significant change. The mean value of the pre-training value was 60% and changed to 85% and 90% in the post training and in their working area. Additionally, based on the gaps identified on the result of the training, consecutive mentoring and follow-up was conducted by senior physicians and public health experts.

Furthermore, the manuscript does not address how decentralization could impact healthcare equity. For example, are rural areas with limited resources at risk of unequal care quality? The manuscript makes a useful contribution to the understanding of decentralized diabetes care in Ethiopia, but its novelty is limited by the replication of methods tested in other LMICs. The small sample size, lack of long-term data, and underexplored systemic challenges weaken the overall impact. Refining the discussion, emphasizing Ethiopia-specific insights, and addressing methodological limitations will significantly enhance the manuscript’s value.

Response: Thank you, you are right it has limitation on the sample size issue and method of participants selection, but we have tried to follow the WHO recommended procedure and assessed the clinical parameters and outcome with the randomly selected participants followed in the Ayder Comprehensive Specialized Hospital. As mentioned in the above the plan was to scale-up its coverage in to other facilities, but due to the war stopped after the pilot. Based on your comment we have tried to address the discussion part.

Reviewer #3:

1. In introduction, paragraph no. 3, line no 3, it has been written as "76% do not even know that they have diabetes", it is suggested not to start the sentence with numerical value.

Response: Thank you very much, you are right sentence should not be start with number and we have revised (Line 58-62)

2. There are no references in the fourth paragraph of introduction.

Response: Thank you, we have put it the reference (line 68)

3. In data collection tool: what type of questionnaire was used (structured, semi-structured...), what were the major contents of data collection tools (written but not clear)?

Response: Thank you, this is an interesting suggestion, we have revised and included the type of data collection tool and its contents. (line 143-148)

4. Result: (Needs to be re-written)

- "Behavioural and comorbidity characteristics section: it is suggested to make the description and figure uniform.

Response: Thank you for your comment, we have revised the table based on the narration in the sub-section under behavioural and comorbid characteristics. Though, we have more information in the table which was narrated and cited in the description. (line 179-191)

- "Clinical symptoms and measurements" section: description has been provided, however, there is no table (Table no. 3 has been mentioned to refer for the result though)

- No figure for comparison related data (only description provided)

Response: Thank you very much for the gap you showed us, we have included a table for the clinical symptoms and measurements section and revised accordingly. (line 194-199 and Table-3)

5. In consent to participate, it is suggested to replace the word 'clients' with participants. It is better if sentences would be added on anonymity, risk if any.

Response: The word clients was replaced by participants, and we have added sentence related to anonymity

Confusion:

- Are the both intervention group and comparison group same?? I mean both patients group? Maybe I have misunderstood.

Response: thank you for your question. Both groups are from different types of facilities. The participants in the implementation group were from Hagereselam primary hospital, which was voluntarily moved from Ayder comprehensive specialized hospital and diagnosed in the implementation site, and was compared with participants other than those included in the implementation group selected from comprehensive specialized hospital that come from a similar setup.

---

## [Decision Letter · Decision Letter 1]

18 Feb 2025

Diabetes service decentralization to primary healthcare unit in Tigray, Ethiopia: A pilot study

PONE-D-24-29698R1

Dear Mr. Tesfahunegn,

We’re pleased to inform you that your manuscript has been judged scientifically suitable for publication and will be formally accepted for publication once it meets all outstanding technical requirements.

Kind regards,

Efrem Kentiba, PhD

Academic Editor

PLOS ONE

Additional Editor Comments (optional):

The order of the authors has changed during the revision process compared to the initially submitted version. You must justify these changes or confirm that all authors have agreed to them by responding to the editorial office or addressing any inquiries related to this change.

Reviewers' comments:

Reviewer's Responses to Questions

**Comments to the Author**

1. If the authors have adequately addressed your comments raised in a previous round of review and you feel that this manuscript is now acceptable for publication, you may indicate that here to bypass the “Comments to the Author” section, enter your conflict of interest statement in the “Confidential to Editor” section, and submit your "Accept" recommendation.

Reviewer #1: All comments have been addressed

Reviewer #3: All comments have been addressed

2. Is the manuscript technically sound, and do the data support the conclusions?

Reviewer #1: Yes

Reviewer #3: Yes

3. Has the statistical analysis been performed appropriately and rigorously? 

Reviewer #1: Yes

Reviewer #3: Yes

4. Have the authors made all data underlying the findings in their manuscript fully available?

Reviewer #1: Yes

Reviewer #3: Yes

5. Is the manuscript presented in an intelligible fashion and written in standard English?

Reviewer #1: Yes

Reviewer #3: Yes

6. Review Comments to the Author

Reviewer #1: The authors have addressed my questions satisfactorily. I don't have any more questions.

Thank you!

Reviewer #3: (No Response)

7. PLOS authors have the option to publish the peer review history of their article (what does this mean? ). If published, this will include your full peer review and any attached files.

**Do you want your identity to be public for this peer review?** For information about this choice, including consent withdrawal, please see our Privacy Policy .

Reviewer #1: **Yes: ** Hale Teka, MD

Reviewer #3: No

---

## [Editor Report · Acceptance letter]

PONE-D-24-29698R1

PLOS ONE

Dear Dr. Nigusse,

I'm pleased to inform you that your manuscript has been deemed suitable for publication in PLOS ONE. Congratulations! Your manuscript is now being handed over to our production team.

Kind regards,

on behalf of

Dr. Efrem Kentiba

Academic Editor

PLOS ONE